# The FLARE Score and Circulating Neutrophils in Patients with Cancer and COVID-19 Disease

**DOI:** 10.3390/cancers16172974

**Published:** 2024-08-26

**Authors:** Elia Seguí, Juan Manuel Torres, Edouard Auclin, David Casadevall, Sara Peiro Carmona, Juan Aguilar-Company, Marta García de Herreros, Teresa Gorría, Juan Carlos Laguna, Marta Rodríguez, Azucena González, Nicolas Epaillard, Javier Gavira, Victor Bolaño, Jose C. Tapia, Marco Tagliamento, Cristina Teixidó, Hugo Arasanz, Sara Pilotto, Rafael Lopez-Castro, Xabier Mielgo-Rubio, Cristina Urbano, Gonzalo Recondo, Mar Diaz Pavon, Maria Virginia Bluthgen, José Nicolas Minatta, Lorena Lupinacci, Fara Brasó-Maristany, Aleix Prat, Alexandru Vlagea, Laura Mezquita

**Affiliations:** 1Translational Genomics and Targeted Therapies in Solid Tumors Group, August Pi i Sunyer Biomedical Research Institute (IDIBAPS), 08036 Barcelona, Spain; segui@recerca.clinic.cat (E.S.); garciadehe@clinic.cat (M.G.d.H.); laguna@clinic.cat (J.C.L.); alprat@clinic.cat (A.P.); 2Department of Medical Oncology, Hospital Clinic of Barcelona, 08036 Barcelona, Spain; tgorria@clinic.cat; 3Department of Immunology, Hospital Clínic de Barcelona, 08036 Barcelona, Spaineagonzal@clinic.cat (A.G.);; 4Department of Medical Oncology, Hopital Europeen George Pompidou, AP-HP, Université Paris Cité, 75015 Paris, France; 5Department of Medical Oncology, Hospital del Mar, 08036 Barcelona, Spain; 6Department of Medical Oncology, Vall d’Hebron University Hospital, 08036 Barcelona, Spain; 7Department of Medical Oncology, Parc Taulí Hospital Universitari, 08208 Sabadell, Spain; 8Department of Medical Oncology, Hospital de la Santa Creu i Sant Pau, 08036 Barcelona, Spain; jgavira@seom.org (J.G.);; 9Department of Internal Medicine and Medical Specialties, University of Genova, 16126 Genova, Italy; tagliamento.marco@gmail.com; 10Academic Medical Oncology Unit, IRCCS Ospedale Policlinico San Martino, 16132 Genova, Italy; 11Department of Pathology and CORE Molecular Biology laboratory, Hospital Clinic of Barcelona, 08036 Barcelona, Spain; 12Faculty of Medicine and Health Sciences, University of Barcelona, 08036 Barcelona, Spain; 13Department of Medical Oncology, Hospital Universitario de Navarra, Instituto de Investigación Sanitaria de Navarra (IdiSNA), 31008 Pamplona, Spain; hugo.arasanz.esteban@navarra.es; 14Section of Innovation Biomedicine—Oncology Area, Department of Engineering for Innovation Medicine (DIMI), University of Verona and University and Hospital Trust (AOUI) of Verona, 37126 Verona, Italy; sara.pilotto@univr.it; 15Department of Medical Oncology, Hospital Clinico Universitario de Valladolid, 47003 Valladolid, Spain; 16Department of Medical Oncology, Hospital Universitario Fundación Alcorcon, 28922 Alcorcon, Spain; xmielgo@hotmail.com; 17Department of Medical Oncology, Hospital General de Granollers, 08402 Granollers, Spain; 18Department of Medical Oncology, Centro de Educación Médica e Investigaciones Clínicas “Norberto Quirno” (CEMIC), Buenos Aires C1000, Argentina; 19Department of Medical Oncology, Hospital Aleman, Buenos Aires C1118AAT, Argentina; 20Department of Medical Oncology, Hospital Italiano de Buenos Aires, Buenos Aires C1199, Argentina; drnicolasminatta@gmail.com (J.N.M.); lorena.lupinacci@gmail.com (L.L.); 21Institute of Oncology (IOB)—Hospital Quirón Salud, 08023 Barcelona, Spain; 22Reveal Genomics, 08036 Barcelona, Spain

**Keywords:** cancer, COVID-19, dNLR, neutrophils, IL-6

## Abstract

**Simple Summary:**

Understanding the inflammatory interplay between cancer and COVID-19 infection is essential for improving patient care. Our study bridges a vital knowledge gap by exploring how a pre-existing tumor-induced inflammatory state can exacerbate the inflammatory response to COVID-19, adversely impacting COVID-19 outcomes. We introduce the FLARE score, a robust predictor derived from circulating inflammatory markers, that provides clinicians a practical tool for early identification of patients with cancer at higher risk of severe COVID-19 complications who might benefit most from immediate and intensive treatment strategies. Additionally, our study also underscore the role of immature neutrophils in the progression of COVID-19 in patients with cancer, advocating for further investigation into how these cells contribute to both cancer and COVID-19 disease.

**Abstract:**

Purpose: Inflammation and neutrophils play a central role in both COVID-19 disease and cancer. We aimed to assess the impact of pre-existing tumor-related inflammation on COVID-19 outcomes in patients with cancer and to elucidate the role of circulating neutrophil subpopulations. Methods: We conducted a multicenter retrospective analysis of 524 patients with cancer and SARS-CoV-2 infection, assessing the relationship between clinical outcomes and circulating inflammatory biomarkers collected before and during COVID-19 infection. Additionally, a single-center prospective cohort study provided data for an exploratory analysis, assessing the immunophenotype of circulating neutrophils and inflammatory cytokines. The primary endpoints were 30-day mortality and the severity of COVID-19 disease. Results: Prior to COVID-19, 25% of patients with cancer exhibited elevated dNLR, which increased to 55% at the time of COVID-19 diagnosis. We developed the FLARE score, incorporating both tumor- and infection-induced inflammation, which categorized patients into four prognostic groups. The poor prognostic group had a 30-day mortality rate of 68%, significantly higher than the 23% in the favorable group (*p* < 0.0001). This score proved to be an independent predictor of early mortality. This prospective analysis revealed a shift towards immature forms of neutrophils and higher IL-6 levels in patients with cancer and severe COVID-19 infection. Conclusions: A pre-existing tumor-induced pro-inflammatory state significantly impacts COVID-19 outcomes in patients with cancer. The FLARE score, derived from circulating inflammatory markers, emerges as an easy-to-use, globally accessible, effective tool for clinicians to identify patients with cancer at heightened risk of severe COVID-19 complications and early mortality who might benefit most from immediate and intensive treatment strategies. Furthermore, our findings underscore the significance of immature neutrophils in the progression of COVID-19 in patients with cancer, advocating for further investigation into how these cells contribute to both cancer and COVID-19 disease.

## 1. Introduction

The coronavirus disease 2019 (COVID-19) pandemic dramatically reshaped oncological care, revealing the heightened vulnerability of patients with cancer who exhibit a high likelihood of severe COVID-19 complications and increased mortality [1,2,3]. However, there is a limited understanding regarding which specific cancer pre-conditions aggravate the course of COVID-19 infection, underscoring the need to identify patients with cancer at risk who require prompt management and close surveillance.

SARS-CoV-2 infection leads to a broad spectrum of disease severities. Immune changes associated with severe disease include a pronounced pro-inflammatory cytokine storm and an expansion of immature myeloid populations, with neutrophils playing a pivotal role [4,5,6,7,8,9,10]. This inflammatory cascade, marked by a suite of indicators such as neutrophilia, the derived neutrophil-to-lymphocyte ratio (dNLR), c-reactive protein (CRP), interleukin-6 (IL-6), and tumor necrosis factor-alpha (TNF-alpha), is essential not only to the innate immune defense against COVID-19, but also in the context of cancer [11,12]. Moreover, neutrophil dysfunction, including the formation of neutrophil extracellular traps (NETs), has been linked to acute respiratory distress syndrome (ARDS), thromboembolic events, and increased mortality in COVID-19 [6], reflecting the adverse outcomes often associated with neutrophils in patients with cancer [12].

This study investigates the underexplored critical influence of pre-existing cancer-induced inflammation on clinical outcomes of COVID-19. It hypothesizes that a cancer-induced pro-inflammatory state may intensify the inflammatory response to SARS-CoV-2, adversely impacting COVID-19 outcomes. The findings may have significant implications for the management strategies for patients with cancer and COVID-19 infection.

## 2. Methods

### 2.1. Study Design, Study Population, and Data Collection

#### 2.1.1. Retrospective Clinical Cohort

We conducted a multi−center, international, retrospective cohort study involving 524 cancer patients with confirmed SARS-CoV-2 infection, identified between March 2020 and December 2020. The inclusion criteria were as follows: (1) age ≥ 18 years; (2) a history of solid tumor malignancy prior to or during the COVID-19 disease course; and (3) SARS-CoV-2 infection confirmed by reverse transcription polymerase chain reaction (RT−PCR) on a nasopharyngeal swab. The cohort included patients with active malignancies and patients in cancer remission. The participating centers are listed in Appendix A. Clinical pseudonymized data, including laboratory and radiologic results, were retrospectively collected from electronic medical reports. The retrospective nature of this study allowed for a waiver of prospective informed consent, with pseudonymized data collected as per standard of care protocols.

#### 2.1.2. Prospective Exploratory Cohort

In parallel, a prospective exploratory cohort study was conducted at Hospital Clínic de Barcelona from July 2020 to May 2021. Patients with cancer newly diagnosed with SARS-CoV-2 infection were included. The eligibility criteria, as well as the clinical, laboratory, and radiological data collected, were the same as those for the retrospective cohort. Informed consent was obtained for prospective blood collection at various timepoints during COVID-19 infection, enabling dynamic immunological assessment. The immunophenotype of circulating immune cells was assessed in fresh peripheral blood at each timepoint using four distinct flow cytometry panels (Appendix A). In parallel, a multiplex cytokine assay (ProcartaPlex Multiplex Cytokine Assay from Thermo Fisher) was applied to measure 20 different cytokines in these same blood samples (Appendix A). Healthy volunteers and cancer patients without SARS-CoV-2 infection also participated as control groups after providing informed consent. The same four flow cytometry panels and cytokine assay were assessed per each sample

#### 2.1.3. Data Collection

The clinical data encompassed patients’ demographics, comorbidities, oncological history, and detailed COVID-19 features. The latter included symptom presentation, radiological findings, length of hospitalization, therapeutic interventions, oxygen requirements, intensive care unit admissions, secondary complications, and 30-day mortality. Laboratory data, specifically circulating inflammatory markers such as dNLR, were collected at two distinct timepoints: pre-SARS-CoV-2 infection (15–45 days prior to diagnosis) and at the time of COVID-19 diagnosis. dNLR was defined as [neutrophils]/[leucocytes-neutrophils]) and was considered high if >3, consistent with previous literature [13].

### 2.2. Endpoints

The primary endpoints were 30-day mortality and the severity of COVID-19 disease: (a) mild COVID-19 was defined as the presence of only mild symptoms without any requirement for supplemental oxygen; (b) moderate COVID-19 was defined as a condition where supplemental oxygen was required but at a flow rate of less than 15 L per minute (L/min); (c) severe COVID-19 was defined as a condition where supplemental oxygen was delivered at a flow rate of 15 L/min or more.

### 2.3. Immunophenotyping

A multiparametric flow cytometry analysis was performed with whole-blood samples collected into EDTA, through the staining of cell surface markers and standard flow cytometry methods. Briefly, 100 µL of heparinized whole blood was incubated for 15 min at room temperature (RT) with the appropriate concentration of monoclonal antibodies (mAbs). The cells were then incubated with 2 mL of BD lysing solution 1× (BD Bioscience, United States) for 15 min at RT to lyse erythrocytes and fix the cells. Finally, the cells were washed two times with phosphate-buffered saline (PBS) 1×. Peripheral blood leukocytes were stained as indicated with the following combinations of antibodies: (1) for enumeration, CD45-APCH7, CD16-BV510, CD3-APC, CD4-FITC, CD8-PercpCy5.5, CD19-PECy7, CD244-BV421, and CD56-PE; (2) neutrophil subsets: CD15-PE, CD14-PercpCy5.5, CD16-PECy7, CD10-BV421, LOX 1 FITC, CD11b-BV510, CD64-APCH7, and CD62L-APC; (3) monocyte subsets: CD16-BV510, CD14-PercpCy5.5, CCR5-BB515, CD44-PECy7, CD71-APCH7, CD33-APC, HLA-DR-V450, and CD163-PE; (4) T lymphocyte subsets: TCRγδ-PE, CD3-APC-H7, CD4-FITC, CD8-PercPCy5.5, CD45RA-PECy7, CCR7-BV510, HLA-DR-APC, and PD-1-BV421. All monoclonal antibodies used in the panels were obtained from BD Biosciences (Franklin Lakes, Bergen County, NJ, USA), except for LOX1, which was sourced from ThermoFisher (Waltham, MA, USA). All samples studied with flow cytometry were acquired using a FACSCanto-II (BD Bioscience) cytometer. A minimum of 200,000 events were acquired for the different populations studied. The FCS files were exported from BD FACSDiva Software in a 3.0 format. The analysis was performed using Infinicyt software (Infinicyt 2.0.6, Cytognos SL, Salamanca, Spain).

### 2.4. Serum Analyses

The quantification of cytokines including sE-Selectin; GM-CSF; ICAM-1/CD54; IFN alpha; IFN gamma; IL-1 alpha; IL-1 beta; IL-4; IL-6; IL-8; IL-10; IL-12p70; IL-13; IL-17A/CTLA-8; IP-10/CXCL10; MCP-1/CCL2; MIP-1alpha/CCL3; MIP-1 beta/CCL4; sP-Selectin; and TNF alpha was performed using the Inflammation 20-plex Human ProcartaPlex-Panel (ThermoFisher). Plasma samples were thawed at RT, diluted 1:2 with Universal Assay Buffer (provided by the manufacturer), and assayed according to the manufacturer’s instructions. Measurements were performed with a Bio-Plex 200 System, and acquisitions and analyses were performed by using Luminex 100/200.

### 2.5. Statistical Analysis

Categorical variables are stated as numbers (*n*) and percentages (%). Continuous variables are shown as medians with interquartile ranges (IQR) unless indicated otherwise.

Group comparisons utilized the Chi−square or Fisher’s exact test for categorical variables and the unpaired T−test, Wilcoxon rank-sum test, or ANOVA for continuous variables. Survival functions were estimated using the Kaplan–Meier method, with the log-rank test used for comparisons. Median follow-up was calculated using the reverse Kaplan–Meier method.

Factors associated with 30-day mortality and COVID-19 severity were tested with logistic regression in univariate and multivariate analyses. Subsets identified from both patients and controls were represented through principal component analysis (PCA) and visualized as heatmaps using the web tool ClustVis.

All *p*-values lower than 0.05 were considered statistically significant. Statistical analyses were performed using R software (Version 2023.06.1+524).

## 3. Results

### 3.1. Study Population

The characteristics of the study population, including 524 patients in the retrospective cohort and 27 in the prospective cohort, are detailed in Appendix A.

#### 3.1.1. Retrospective Clinical Cohort

In the retrospective cohort, the median duration of follow-ups was 84 days (95% CI 78–90). Patients were predominantly male (52%), with a median age of 69 years (range: 35–98 years). Hypertension was present in 49%, while 20% had cardiovascular disease. With respect to cancer, 70% had active disease, 64% were at an advanced stage, and 78% had a baseline Eastern Cooperative Oncology Group (ECOG) Performance Status (PS) of less than or equal to 1. The most common cancer types were thoracic (26%), gastrointestinal (24%), breast (19%), and genitourinary (14%). Most patients (57%) were under systemic therapy, including chemotherapy (62%), endocrine therapy (22%), and immunotherapy (14%). At the time of COVID-19 diagnosis, 41% presented with moderate to severe symptoms, with fever (68%), cough (53%), and dyspnea (48%) being the most common ones. The majority (90%) required hospitalization, with a median length of stay of 13.5 days (range 1–73). Intermediate or intensive care was necessary for 12% of patients. Patients received treatment with antibiotics (74%), hydroxychloroquine (65%), antiviral therapy (31%), corticosteroids (22%), and other immunomodulatory drugs (10%). Overall mortality was 29%, with a 25% incidence of severe acute respiratory failure (SARF). In those requiring higher levels of care, mortality was 48%, and SARF was 90%. Mortality rates correlated with baseline PS, with 19% for PS ≤ 1 and 51% for PS ≥ 2 (*p* < 0.001).

#### 3.1.2. Prospective Exploratory Cohort

In the prospective cohort, patients were predominantly female (52%), with a median age of 65 years (range: 49–82 years). Hypertension was reported in 52% of patients and cardiovascular disease in 15%. With respect to cancer, 96% had active disease, with 82% at an advanced stage, and 56% had a baseline PS of less than or equal to 1. The most frequent cancer types included genitourinary (26%), gastrointestinal (22%), thoracic (19%), and breast (15%). Most patients (89%) were undergoing systemic therapy, predominantly chemotherapy (67%) and immunotherapy (15%). At the onset of COVID-19, 70% had moderate to severe symptoms, with fever (78%), cough (37%), and dyspnea (33%) as common presentations. All patients were hospitalized, with a median stay of 17 days (range 3–71). Intermediate or intensive care was required for 15% of patients. The treatment regimen included antibiotics (67%), antiviral drugs (56%), corticosteroids (56%), plasma therapy (26%), and other immunomodulatory drugs (33%). The overall mortality rate stood at 15%, with an 18% rate of SARF. In the subset admitted to higher levels of care, mortality and SARF rates were 50% and 75%, respectively. Mortality rates varied with baseline PS, with 0% for PS ≤1 and 33% for PS ≥ 2 (*p* = 0.057).

### 3.2. dNLR in Patients with Cancer and COVID-19 Infection

In our retrospective clinical cohort of patients with cancer and COVID-19 infection, a pre-infection pro-inflammatory status, indicated by an elevated baseline dNLR (>3), was observed in 25% of the patients. At the timepoint of COVID-19 diagnosis, the incidence of high dNLR had increased to 55%, suggesting an increase in inflammatory response upon viral infection. The change in dNLR, referred to as delta dNLR (△dNLR), represents the percentage increase from the baseline to the COVID-19 diagnosis timepoint. The median delta dNLR was +70% (IQR: 0–349%), with an average increase of +100%. Notably, 42% of patients experienced a doubling (100% increase) of dNLR, as depicted in Figure 1A, indicative of a significant inflammatory response associated with COVID-19 infection (Figure 1A).

### 3.3. Building of the FLARE Score

To develop the FLARE score, we first established criteria for tumor-induced pro-inflammatory status (T+) and COVID-induced pro-inflammatory status (I+). Tumor-induced pro-inflammatory status (T+) was defined as a high dNLR at baseline, prior to COVID-19 infection, indicating a pre-existing inflammatory condition due to the tumor. COVID-induced pro-inflammatory status (I+) was defined as a 100% increase in dNLR between baseline and the time of COVID-19 diagnosis, indicating an inflammatory response triggered by the COVID-19 infection (Figure 1B).

By combining both tumor-induced and COVID-induced inflammation, the FLARE score facilitated the stratification of patients into four subgroups based on their inflammation status: (1) T+/I+ (poor), indicating both inflammation (high dNLR) at baseline prior to COVID-19 diagnosis related to the tumor and an additional ≥100% increase in dNLR triggered by COVID-19 infection; (2) T+/I− (T−only), indicating only inflammation (high dNLR) at baseline related to the tumor without a significative increase in dNLR related to COVID-19; (3) T−/I+ (I−only), indicating no tumor-related inflammation prior to COVID-19 but a ≥100% increase in dNLR due to COVID-19 infection; and (4) T−/I− (favorable), indicating the absence of inflammation, with no high dNLR at baseline or increase related to COVID-19.

The distribution of patients according to the FLARE score categories was as follows: 5% were categorized as poor (*n* = 19), 20% as T−only (*n* = 74), 37% as I−only (*n* = 136), and 38% as favorable (*n* = 140).

The median dNLR at baseline was highest in the T + /I− (T−only) group [4.26 (IQR: 3.77–6.86)], followed by the T + /I+ (poor) [3.88 (IQR: 3.42–4.38)], the T−/I− (favorable) [1.74 (IQR: 1.24–2.34)], and the T−/I+ (I−only) [1.41 (IQR: 0.97–1.96)] (*p* < 0.001) groups. The median delta dNLR showed the greatest increase in the T + /I+ (poor) group [7.46 (IQR: 5.31–9.9)], followed by the T−/I+ (I−only) [3.56 (IQR: 2–5.77)], the T−/I− (favorable) [0.35 (IQR: −0.24–0.81)], and the T + /I (T−only) groups [−0.36 (IQR: −2.49–1.57)] (*p* < 0.001). In 43% of patients within the T−only group and 62% of patients within the favorable group, some increase in dNLR following COVID-19 infection was observed, but it did not exceed the cutoff of a 100% increase

Significant differences were observed across the FLARE groups in terms of active disease at diagnosis, stage IV disease, and known circulatory inflammatory biomarkers such as platelet count, LDH levels, and albumin levels, among others. These and other characteristics of patients within these FLARE categories are summarized in Table 1A.

### 3.4. The FLARE Score Is Correlated with Early Mortality

Table 1B summarizes COVID-19 outcomes across FLARE groups. Although this analysis is limited by small sample sizes and potential confounding factors, no significant differences were observed in ICU admissions across the FLARE subgroups. Similarly, despite a trend, no significant differences were found in the rates of severe acute respiratory failure (*p* = 0.275).

However, significant differences were evident when considering all COVID-19 complications (including acute respiratory failure, acute kidney injury, acute liver injury, acute cardiac injury, secondary infections, thromboembolic events, and cryptogenic organizing pneumonia). Complications were most prevalent in the T+/I+ (poor) group at 88%, followed by 79% in the T+/I− (T−only) group, 75% in the T−/I+ (I−only) group, and 54% in the T−/I− (favorable) group (*p* < 0.001).

The 30-day overall mortality rate was 29%. A gradient in mortality rates was observed across the FLARE groups: the T+/I+ (poor) group experienced the highest mortality at 68%, followed by the T+/I− (T−only) group at 39%, the T−/I+ (I−only) group at 33%, and the T−/I− (favorable) group at 23% (*p* < 0.001) (Table 1B and Figure 2A). This stratification indicates a clear correlation between pro-inflammatory status and COVID-19 outcomes. A multivariate analysis, adjusting for variables including age, gender, comorbidities, tumor stage, and PS, confirmed the poor FLARE group as an independent predictor of mortality at 30 days, with an odds ratio (OR) of 5.91 (95% CI: 1.83–19.3; *p* = 0.003) (Figure 2B).

Though limited by smaller numbers, early mortality according to FLARE was further studied in the four main cancer types (breast, n = 124; genitourinary, n = 71; gastrointestinal, n = 124; thoracic cancers, n = 134). Despite differences in mortality across cancer types within the different FLARE groups, early mortality was consistently higher, except for breast, in the poor FLARE group. In breast cancer, the early mortality rates were 10.7% in T−/I−, 10.5% in T−/I+, 50% in T+/I−, and 33.3% in T+/I+ (*p* = 0.07) (Kaplan–Meier Curves in Appendix A, *p* = 0.009). In genitourinary cancer, the early mortality rates were 50% in T−/I−, 23.5% in T−/I+, 11% in T+/I−, and 100% in T+/I+ (*p* = 0.018) (Kaplan–Meier Curves in Appendix A, *p* = 0.007). In gastrointestinal cancer, the early mortality rates were 18.9% in T−/I−, 41.18% in T−/I+, 25% in T+/I−, and 80% in T+/I+ (*p* = 0.02) (Kaplan–Meier Curves in Appendix A, *p* = 0.01). In thoracic cancers, the early mortality rates were 29.6% in T−/I−, 41.7% in T−/I+, 48% in T+/I−, and 60% in T+/I+ (*p* = 0.4) (Kaplan–Meier Curves in Appendix A, *p* = 0.004).

### 3.5. Circulating Neutrophils in Patients with Cancer and COVID-19 Infection

In the prospective exploratory cohort, the immunophenotype of circulating immune cells was examined in 27 cancer patients at different timepoints of COVID-19 infection, with a total of 37 samples analyzed. The patients’ characteristics in this exploratory cohort are described in Appendix A. This group was compared to a control group of 10 patients with cancer without COVID-19 infection (cancer control, CC) and 6 healthy volunteers (HV). The severity of COVID-19 in these 37 samples varied, and the distribution was as follows: 16% baseline/asymptomatic (n = 6; COVID-19 diagnosis but with no symptoms), 22% mild (n = 8, COVID-19 symptoms but no oxygen requirements), 22% moderate (n = 8, oxygen requirements < 15 mL/min), and 30% severe (n = 11, oxygen requirements > 15 mL/min) and in the recovery phase post−infection (n = 4).

Significant differences in various immune cell populations, including lymphocytes, monocytes, eosinophils, and macrophages, were identified between the different COVID-19 severity groups, cancer controls, and healthy volunteers. Of particular interest were the circulating neutrophils. Patients with cancer had a median neutrophil percentage of 56.7% (39–78.4%), which was higher than that of healthy volunteers at 35.8% (25.6–45%). This percentage increased dramatically in patients with both cancer and COVID-19 infection to 73.9% (66.7–88.3%) (*p* = 0.0003), as shown in Figure 3A. The increase was even more pronounced in patients with cancer and severe COVID-19 infection, where the median neutrophil percentage escalated to 88.4% (84.5–95.8%).

Further principal component analysis (PCA) differentiated the immunophenotypes of circulating neutrophils in patients with cancer and COVID-19 infection from those in CC and HV, as depicted in Figure 3B. Interestingly, cancer patients across all COVID-19 severity categories—mild, moderate, and severe—tended to cluster together in the PCA plot.

### 3.6. Circulating Immature Neutrophils in Patients with Cancer and COVID-19 Infection

To assess the relevance of specific neutrophil subpopulations in patients with cancer and severe COVID-19 infection, we focused particularly on immature neutrophils due to their known aggressiveness and association with severe COVID-19 infection. These immature neutrophils were characterized as CD10-negative/CD16-positive, with the additional markers CD11b and CD15 expressed variably (CD11bpos/neg, CD15pos/neg), whereas mature neutrophils were identified as CD10-positive/CD16-positive, as detailed in Appendix A. PCA did not reveal distinct profiles in immature neutrophils that correlated with COVID-19 severity (Appendix A). However, a heatmap analysis of immature and mature neutrophil subpopulation exposed distinct clusters, differentiating primarily the severe and moderate COVID-19 cases from the milder forms (Figure 3C). Notably, patients with cancer and severe COVID-19 infection had a higher median percentage of circulating immature neutrophils (4.64% [3.02–10.1%]) compared to those with asymptomatic/mild (2.1% [0.51–2.82%]) or moderate disease (1.42% [0.92–2.43%]) (*p* = 0.360) (Figure 3D). Yet, no definitive link was established between levels of immature neutrophils and 30-day mortality, although this observation was limited by the small number of events (n = 3 for immature neutrophil data).

### 3.7. Correlation between Circulating Inflammatory Cytokines, Circulating Immature Neutrophils, and Circulating Inflammatory Markers (dNLR) in Patients with Cancer and COVID-19 Infection

In our investigation, we utilized a multiplex cytokine assay to measure twenty key circulating inflammatory cytokines, analyzing their levels in relation to the proportion of immature neutrophils and established circulating inflammatory biomarkers such as dNLR. Among the cytokines examined, IL-6 stood out, showing the most pronounced variations across COVID-19 severity groups (Figure 3F), with a discernible trend when comparing patients with cancer and COVID-19 to both CC and HV (Figure 3E). Patients with a higher proportion of immature neutrophils also demonstrated elevated IL-6 levels (Figure 3G), and a robust correlation was observed between IL-6 concentrations and dNLR (corr = 0.705, *p* < 0.001). Details of the patient characteristics from the exploratory cohort—including dNLR, the proportion of immature neutrophils, IL-6 levels, and COVID-19 outcomes—are presented in Table 2, enabling a comprehensive visual assessment of the correlations. However, it is important to note that no association was found between IL-6 levels and 30-day mortality, an observation that was limited by the small number of events (n = 3 for IL-6 data).

### 3.8. Monitoring of Circulating Neutrophils and IL-6

Lastly, we tracked the evolution of circulating neutrophils in patients with sequential samples available throughout the evolution of COVID-19 infection. This dynamic analysis allowed us to discern patterns and highlighted how the emergence of immature neutrophils during COVID-19 evolution could lead to the development of severe COVID-19 disease. Notably, we observed a consistent trend towards a higher proportion of immature neutrophils, alongside elevated IL-6 levels, in samples collected during periods of clinical deterioration compared to those from the baseline or recovery phases, as depicted in Figure 4A,B. Additionally, a case study of FLARE #15 is presented in Figure 4C, which graphically depicts the evolution of circulating immature neutrophils, illustrating the potential impact of these cells on the severity of the patient’s COVID-19 infection.

## 4. Discussion

The interplay between cancer-associated conditions and COVID-19 outcomes has been extensively studied, with factors like comorbidity burden, gender, age, and tumor stage being widely recognized as influential [1,2,3]. Yet, the major role of systemic inflammation, a pathogenic mechanism shared by both cancer progression and COVID-19, remains to be fully elucidated.

Building upon the findings of Dettorre et al. and Cortellini et al. [14,15], which underscore the prognostic value of inflammatory markers such as the neutrophil-to-lymphocyte ratio (NLR) in predicting adverse outcomes and sequelae for patients with cancer and COVID-19 infection, our study aimed to expand the scope by examining the impact of a pre-existing tumor-induced inflammatory state. We explored whether such a state predisposes patients with cancer to an exacerbated innate immune response upon COVID-19 infection.

Numerous routine blood parameters have been investigated as potential circulating inflammatory biomarkers in patients with cancer, including elevated neutrophils, platelets, LDH, and hypoalbuminemia, all of them associated with poor outcomes [16,17]. More recently, novel neutrophil-based ratios, such as the NLR and dNLR, have also been examined. The NLR is a well-known prognostic factor in patients with cancer [11] and has also been shown to be an independent risk factor for severe COVID-19 disease [8]. We focused on the dNLR due to its broader inclusion of granulocyte and monocyte subpopulations and because it offers a relative ratio of neutrophils while considering the total white blood cell count, thus providing a more comprehensive inflammatory profile. High dNLR has been linked to poorer outcomes across various cancer types [18,19,20].

In the present work, we demonstrate how the FLARE score, derived from the dNLR at two different timepoints (prior to COVID-19 diagnosis and at COVID-19 diagnosis), helps stratify patients with cancer and COVID-19 infection into four distinct groups by combining the assessment of both tumor-induced and COVID-induced inflammation. Among our cohort of 524 patients, the 30-day mortality rate was 29%, in line with previous reports [1,2]. Notably, patients with both tumor-related inflammation prior to COVID-19 and additional inflammation triggered by COVID-19, classified as the T+/I+ or poor FLARE group, represented only 5% of patients but had a significantly higher 30-day mortality rate of 68%. In comparison, patients in the T−only, I−only, and favorable FLARE groups had early mortality rates of 39%, 33%, and 23%, respectively (*p* < 0.001). This score proved to be an independent early mortality predictor, even after adjusting for factors such as age, gender, comorbidities, tumor stage, and baseline PS. In addition, when further studied with a stratified analysis on the four main cancer type groups, early mortality was consistently higher, except for breast cancer, in the poor FLARE group. Furthermore, the rate of COVID-19 complications also varied significantly across these groups, with 88% for poor FLARE vs. 79%, 75%, and 54% in T−only, I−only, and favorable FLARE, respectively (*p* < 0.001).

Altogether, our results establish the FLARE score as an attractive, easy-to-use, and inexpensive predictor of both mortality and morbidity in patients with cancer and COVID-19 infection. It can be calculated using values from routine complete blood cell counts taken at just two timepoints, making it easy to implement in various healthcare settings. Thus, the FLARE score emerges as a practical tool for clinicians worldwide, enabling the early identification of patients at higher risk of severe COVID-19 complications and early mortality, and facilitating pragmatic decisions about which patients should receive intensified treatment strategies.

In our exploratory analysis, we examined the immunophenotype of circulating neutrophils in a small cohort of patients with cancer and COVID-19 infection, shedding light on the potential key subpopulations driving the innate immune response to SARS-CoV-2. While neutrophils have historically been viewed as homogenous, recent studies underscore their functional diversity [21,22]. Immature neutrophils are thought to result from premature release from the bone marrow and are known for their altered functional capacity. Their prevalence has been linked to systemic inflammatory responses and clinical deterioration in patients with sepsis and may also contribute to cancer progression [23,24,25]. These immature forms can often be increased in the peripheral blood of patients with cancer due to systemic chemokines produced by either the tumor or cancer treatments [12].

Emerging studies suggest that the shift towards immature forms, with an hyperactivation of immature CD10- subpopulations, may be a key determinant of COVID-19 severity [26,27,28]. In line with other studies, our results confirm a greater number of circulating neutrophils with a different immunophenotype in patients with cancer and COVID-19 compared with cancer controls and healthy volunteers, and a transition towards immature forms in patients who developed severe COVID-19 disease. To the best of our knowledge, this is the first study to confirm this trend towards immature forms in patients with cancer and COVID-19 infection. Although our understanding of immature neutrophils remains incomplete, this evidence warrants further research into how these cells contribute to both cancer and COVID-19 disease.

Research has shown that severe COVID-19 can be predicted not only by emergent myelopoiesis, as previously described, but also by increased levels of inflammatory biomarkers, notably IL-6 [29]. IL-6 is a cytokine with pleiotropic activities and an essential role in downstream signal amplification. It either attaches to its respective cell receptor (IL-6R) or the soluble receptor (sIL-6R), activating both the NF-ĸB and JAK/STAT pathways, which can induce a cytokine storm. This results in the upregulation of pro-inflammatory pathways, downregulation of anti−inflammatory pathways, thrombopoiesis and hypercoagulability, the production of acute phase reactants from the liver, increased vascular permeability, and the recruitment of monocytes and neutrophils. High IL-6 levels are known to predict worse outcomes in COVID-19 [30,31] and contribute to the unique pattern of immune dysregulation seen in severe cases [32]. In patients with cancer, IL-6 is recognized for its role in promoting tumor growth and treatment resistance [33,34,35].

In our study, we analyzed twenty inflammatory cytokines in patients with cancer and COVID-19 and found that IL-6 stood out due to showing the most significant variations across the spectrum of COVID-19 severity. Notably, IL-6 levels were elevated in patients with a higher proportion of immature neutrophils, highlighting the link with an intensified innate immune response and an increase in the recruitment of immature forms. Interestingly, we also found a significant correlation between IL-6 levels and dNLR, used to build the FLARE score. This reinforces the use of dNLR and the FLARE score as biomarkers for tracking the inflammatory trajectory of COVID-19 in patients with cancer.

The dual role of IL-6, both as an indicator of tumor-related inflammation and a predictor of COVID-19 adverse outcomes, suggests that targeting the IL-6 pathway might offer therapeutic benefits in this patient population. Tocilizumab, a recombinant humanized monoclonal antibody directed against the IL-6 receptor, has shown a mortality benefit in severe cases of COVID-19 [36,37], underscoring the potential of IL-6 as a therapeutic target in patients with cancer and COVID-19.

Our study faces several limitations that must be acknowledged. Firstly, the retrospective nature of the multicentric cohort introduces inherent biases, including the potential for missing data and variability in treatment decisions across different centers. This variability may affect the consistency and generalizability of our findings. Secondly, the prospective exploratory cohort was small and involved a limited panel of surface markers for immunophenotyping. This restricts the robustness of our immunophenotyping results and the strength of our conclusions.

Another significant limitation is the uncertainty regarding whether the prognostic value of the FLARE score is specific to COVID-19 or if it could also apply to other viral or bacterial infections. Our study has not yet explored the applicability of the FLARE score beyond the context of COVID-19, leaving open the question of its potential utility in predicting outcomes for patients with cancer affected by other types of infection.

Additionally, our study lacked diversity in terms of patient demographics and some clinical characteristics, which may limit the applicability of our findings to broader patient populations. Furthermore, potential confounding factors, including differences in healthcare access and quality, were not fully accounted for in our analysis.

To address these limitations, we emphasize the need for future studies to validate the FLARE score in larger, more diverse patient populations and to explore its applicability to other infections. Prospective studies with comprehensive data collection will be particularly valuable in confirming the utility and reliability of the FLARE score as a predictive tool. While our study focused only on immunophenotyping and cytokine analysis, incorporating gene expression analysis into future research will be crucial to further our understanding of immature neutrophils and other components of the innate immune response.

Despite these limitations, our findings highlight the importance of a pre-existing tumor-induced pro-inflammatory status, introducing the FLARE score as a practical outcome predictor for patients with cancer and COVID-19 infection. Specifically, our findings regarding the role of immature neutrophils and their potential as prognostic biomarkers underscore the need for further investigation to fully understand their impact on disease progression in both cancer and COVID-19 infection.

## 5. Conclusions

Our study aimed to identify circulating inflammatory biomarkers that are predictive of progression to severe COVID-19 among patients with cancer markers that could potentially be identified before the disease reaches its peak, thereby enabling the initiation of timely therapeutic interventions. Through a retrospective analysis, we discovered that a pre-existing tumor-induced inflammatory state, indicated by elevated dNLR levels prior to the diagnosis of COVID-19, was indicative of adverse outcomes in these patients.

We developed the FLARE score, an integrative measure reflecting inflammation driven by both the tumor and COVID-19 infection. The practical significance of the FLARE score lies in its ability to be calculated using routine blood test values at just two timepoints, making it suitable for implementation in various healthcare settings, from tertiary care centers to community hospitals. Thus, the FLARE score emerges as an easy-to-use and globally accessible tool that effectively stratifies patients with cancer by their risk of COVID-19 complications and early mortality. This tool can aid clinicians in identifying and prioritizing those who might benefit most from immediate and intensive treatment strategies. By guiding clinical decision making, this tool has the potential to enhance patient care and improve survival in this vulnerable population.

In our prospective exploratory cohort, the immunophenotypic analysis of circulating neutrophils revealed that the accumulation of immature neutrophils could be associated with the unfavorable progression of COVID-19 disease. This finding indicates the need to conduct more in-depth characterization of neutrophil subpopulations and improve our ability to predict outcomes for patients with cancer and COVID-19 infection.

Future research is warranted to validate the FLARE score across diverse patient populations and explore its applicability to other infectious diseases, thereby broadening its clinical utility and impact. Additionally, further investigation is needed to understand the role of immature neutrophils not only in infectious diseases but also in cancer progression.

## Figures and Tables

**Figure 1 cancers-16-02974-f001:**
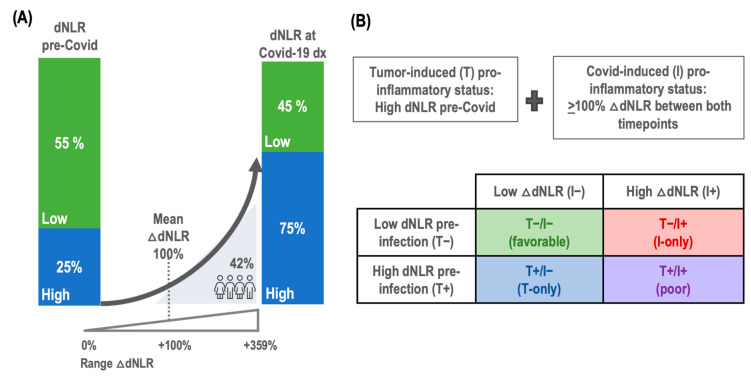
(**A**) dNLR in patients with cancer and COVID-19 infection; (**B**) building of the FLARE score. Legend: dNLR: derived neutrophil-to-lymphocyte ratio; COVID-19 dx: COVID-19 diagnosis; ΔdNLR: delta dNLR.

**Figure 2 cancers-16-02974-f002:**
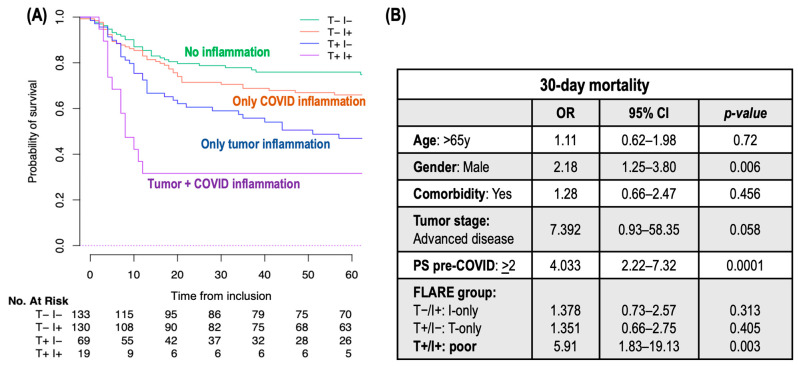
(**A**) Kaplan–Meier curves demonstrating survival across FLARE groups; (**B**) multivariate logistic Regression analysis for 30-Day mortality. Legend: T+/I+: concurrent tumor and infection-related inflammation (poor FLARE group); T+/I−: tumor-related inflammation only (T−only FLARE group); T/I+: COVID-related inflammation only (I−only FLARE group); T−/I−: no inflammation (favorable FLARE group); OR: odds ratio; CI: confidence interval; PS: performance status.

**Figure 3 cancers-16-02974-f003:**
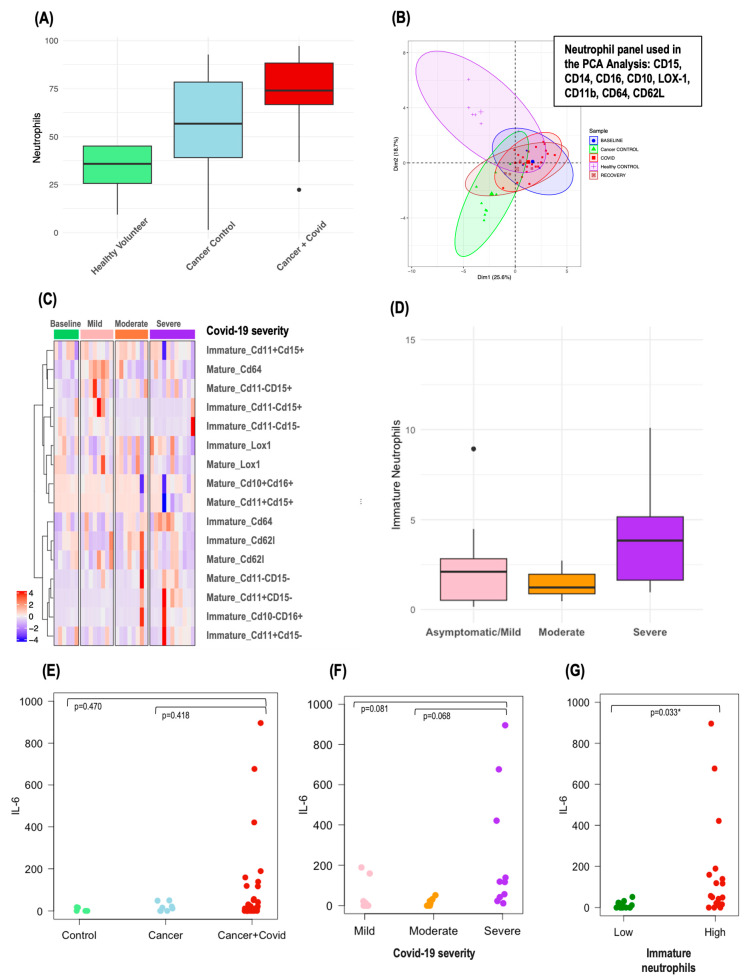
Circulating neutrophils in patients with cancer and COVID-19 infection. Legend: (**A**) Median circulating neutrophils in healthy volunteers (HV), patients with cancer (CC), and patients with cancer and COVID-19 disease (Cancer + COVID); (**B**) principal component analysis (PCA) of circulating neutrophils in HV, CC, and Cancer + COVID; (**C**) heatmap displaying distinct subpopulations of circulating neutrophils in Cancer + COVID patients; (**D**) circulating immature neutrophils based on COVID-19 severity; (**E**) circulating IL-6 levels in HV, CC, and Cancer + COVID; IL-6 levels (**F**) based on COVID-19 disease severity and (**G**) based on circulating immature neutrophils (low vs. high).

**Figure 4 cancers-16-02974-f004:**
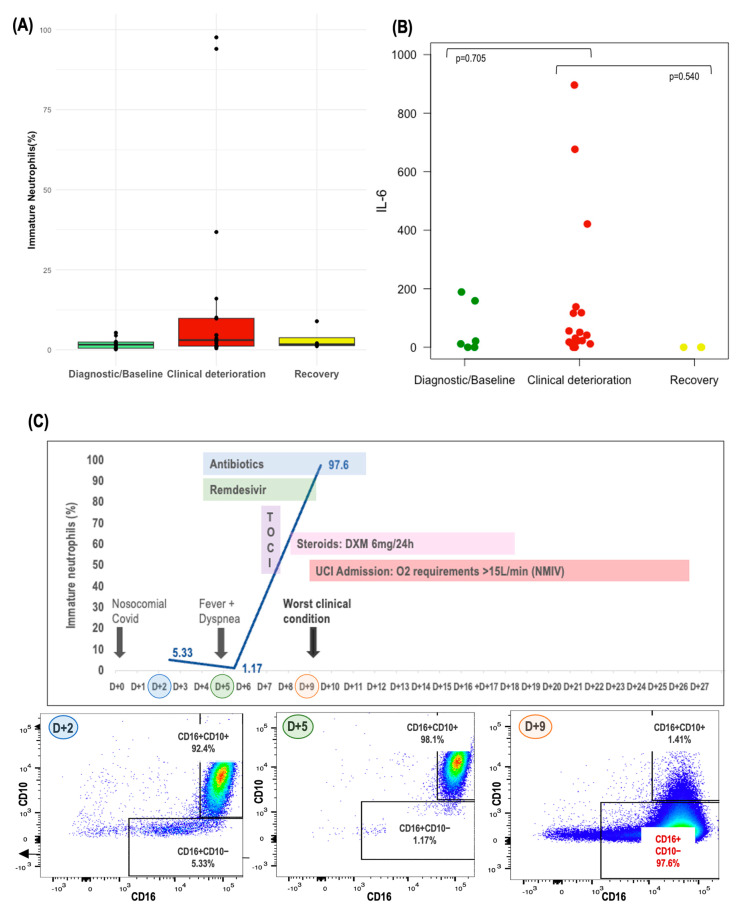
Monitoring of circulating immature neutrophils and IL-6 across COVID-19 infection. Legend: (**A**) Circulating immature neutrophils and (**B**) circulating IL-6 levels across different timepoints during COVID-19 infection. (**C**) Illustration demonstrating the acquisition of immature neutrophils during the course of COVID-19 evolution in FLARE#15 and its association with the development of severe disease. Toci: tocilizumab; DXM: dexamethasone; ICU: intensive care unit; O_2_; oxygen; NIV; noninvasive ventilation.

**Table 1 cancers-16-02974-t001:** (A). Patients’ characteristics by FLARE group. (B). COVID-19 outcomes across FLARE groups. Legend: T+/I+: concurrent tumor and infection-related inflammation (poor FLARE group); T+/I−: tumor-related inflammation only (T−only FLARE group); T/I+: COVID-related inflammation only (I−only FLARE group); T−/I−: no inflammation (favorable FLARE group); ECOG PS: Eastern Cooperative Oncology Group Performance Status; ICU: intensive care unit.

**(A)**					
	**FLARE T−/I−**	**FLARE T−/I+**	**FLARE T+/I−**	**FLARE T+/I+**	***p*-Value**
	**(n = 140)**	**(n = 136)**	**(n = 74)**	**(n = 19)**	
Age (median; range)	68 (39–93)	69 (41–96)	69 (37–88)	69 (43–82)	0.852
Gender: male	60 (43%)	90 (66%)	41 (55%)	12 (63%)	0.001
Current or former smoker	73 (53%)	86 (65%)	44 (62%)	7 (44%)	---
ECOG PS ≤1	111 (80%)	103 (82%)	44 (62%)	14 (78%)	<0.001
Comorbidity					
Hypertension	67 (48%)	60 (44%)	43 (60%)	11 (58%)	0.166
Cardiovascular	33 (24%)	24 (18%)	16 (22%)	5 (26%)	0.613
Cancer type					
Thoracic	31 (22%)	40 (30%)	28 (38%)	5 (26%)	---
Genitourinary	15 (11%)	17 (13%)	9 (12%)	3 (16%)	---
Gastrointestinal	37 (27%)	33 (25%)	15 (20%)	5 (26%)	---
Breast	30 (22%)	20 (15%)	7 (9%)	3 (16%)	---
Active tumor at diagnosis	98 (70%)	103 (75.7%)	62 (83.8%)	19 (100%)	0.005
Advanced stage	88 (63%)	94 (69%)	62 (84%)	18 (95%)	---
Systemic therapy	89 (64%)	83 (61%)	45 (62%)	15 (79%)	0.518
Chemotherapy	59 (54%)	47 (55%)	28 (58%)	13 (87%)	0.143
Immunotherapy	17 (19%)	12 (14%)	9 (19%)	1 (7%)	0.637
Circulating inflammatory biomarkers prior to COVID-19 diagnosis					
Platelets (10^9^/L) (median, IQR)	243 (166.8–292.8)	223.5 (188–295.25)	268 (194–423)	280 (209.5–313.5)	0.011
LDH (U/L)	196 (171.5–265)	202 [167;279]	239 (183.5–353)	196 (171.5–265)	0.025
(median, IQR)					
Albumin (g/L)	41 (38–43)	41 (36–43)	35.7 (32–39)	35.4 (30.7–38.5)	0.001
(median, IQR)					
COVID-19 symptoms at diagnosis					
Fever	97 (69%)	97 (71%)	47 (64%)	13 (68%)	0.782
Cough	80 (58%)	68 (50%)	39 (53%)	8 (42%)	0.494
Dyspnea	59 (42%)	64 (47%)	40 (54%)	12 (63%)	0.207
COVID-19 treatment					
Antibiotics	99 (73%)	115 (86%)	59 (80%)	16 (89%)	0.045
Antiviral therapy	29 (31%)	37 (38%)	18 (32%)	4 (25%)	0.21
Steroids	32 (24%)	27 (21%)	28 (38%)	8 (44%)	---
Immunomodulators	7 (7%)	12 (13%)	5 (9%)	4 (31%)	0.081
Median hospital stay duration (range)	18.5 (4–57)	11.5 (2–44)	18 (2–73)	6 (2–23)	0.177
**(B)**					
	**FLARE T−/I−**	**FLARE T−/I+**	**FLARE T+/I−**	**FLARE T+/I+**	***p*-Value**
	**(n = 140)**	**(n = 136)**	**(n = 74)**	**(n = 19)**	
Admission to ICU	7 (8 %)	11 (14%)	7 (15%)	2 (18%)	0.425
Severe acute respiratory failure	25 (18%)	36 (27%)	18 (24%)	no	0.275
COVID-19 complications	67 (54%)	91 (75%)	53 (79%)	14 (88%)	<0.001
30-day mortality	29 (23%)	42 (33%)	27 (39%)	13 (68%)	<0.001

**Table 2 cancers-16-02974-t002:** Exploratory cohort patient characteristics and COVID-19 outcomes.

FLARE #	Age	Gender	Cancer-Type	Stage	ECOG PS	Days Since Diagnosis	dNLR	Immature Neutrophils (%)	IL-6	COVID-19 Severity	30-Day Mortality
FLARE #1	66	Female	Colorectal	IV	2	D + 6	2.66	-	-	Moderate	No
FLARE #2	60	Male	Prostate	IV	1	D + 10	4.63	-	-	Moderate	No
FLARE #3	82	Male	Prostate	IV	1	D + 3	0.19	-	-	Moderate	No
FLARE #4	61	Male	Bladder	IV	2	D + 15	10.73	-	-	Severe	Yes
FLARE #5	49	Male	Suprarenal	IV	0	D + 3	2.31	1.03	0	Moderate	No
FLARE #6	65	Female	Breast	IV	1	D + 2	2.59	2.72	21.93	Severe	No
						D + 6	2.92	3.09	41.37		
FLARE #7	66	Female	Lung	IV	3	D + 3	18.69	1.15	22.76	Severe	Yes
FLARE #8	59	Male	Esophagus	IV	1	D + 3	0.47	0.15	-	As/Mild	No
							2.21	0.59	-		
FLARE #9	57	Male	Colorectal	IV	0	D + 5	0.27	4.48	21.04	As/Mild	No
FLARE #10	77	Female	Gastric	III	2	D + 9	6.42	0.23	-	As/Mild	No
FLARE #11	56	Female	Breast	IV	2	D + 2	8.49	0.71	-	Moderate	No
						D + 9	12.4	2.14	-		
						D + 16	12.53	1.4	0		
FLARE #12	55	Female	Colorectal	IV	1	D + 1	3.21	2.19	188.85	As/Mild	No
FLARE #13	74	Female	Lung	II	2	D + 1	1.97	2.01	0	As/Mild	No
						D + 8	1.72	-	-		
FLARE #14	61	Female	Ovarian	IV	1	D + 8	1.41	2.39	158.69	As/Mild	No
FLARE #15	60	Male	Bladder	IV	2	D + 2	7.72	5.33	-	Severe	No
						D + 5	10.88	1.17	50.4		
						D + 9	49.13	97.6	895.93		
FLARE #16	58	Male	Head and Neck	III	1	D + 3	2.74	3.64	0	As/Mild	No
						D + 7	2.55	8.93	00		
						D + 19	3.31	-	-		
FLARE #17	73	Male	Esophagus	IV	3	D + 4	7.96	4.58	676.46	Severe	Yes
FLARE #18	71	Male	Thyroid	IV	1	D + 4	2.61	1.42	11.43	Moderate	No
FLARE #19	54	Female	Lung	IV	1	D + 15	2.69	0.82	0	Moderate	No
						D + 46	3.28	1.12	-		
FLARE #20	77	Female	Breast	IV	3	D + 16	2.77	9.75	116.25	Severe	No
						D + 23	6.32	4.64	55.676		
						D + 25	6.32	3.02	138.22		
FLARE #21	72	Female	Lung	IV	1	D + 4	1.22	94	17.83	Moderate	No
FLARE #22	55	Female	Ovarian	IV	1	D + 3	1.31	0.28	11.43	As/Mild	No
FLARE #23	73	Female	Head and Neck	III	3	D + 8	10.52	16	118.09	Severe	No
						D + 10	4.64	10.1	421.18		
						D + 17	3.95	-	-		
FLARE #24	50	Female	Breast	IV	1	D + 12	2.83	1.23	0	As/Mild	No
						D + 45	1.18	2.55	-		
FLARE #25	62	Male	Head and Neck	III	1	D + 3	12.86	36.8	-	Severe	No
						D + 10	4.11	2.08	0		
FLARE #27	81	Male	Lung	IV	2	D + 4	3.91	0.96	11.43	Severe	Yes
FLARE #28	65	Female	Renal	IV	0	D + 8	0.74	-	-	As/Mild	No
FLARE #29	72	Male	Prostate	IV	2	D + 2	6.59	0.46	31.07	Moderate	No

**Legend**: ECOG PS: Eastern Cooperative Oncology Group Performance Status; dNLR: derived neutrophil-to-lymphocyte ratio (highlighted if >5.76 (upper tertile)); IMN: immature neutrophils (highlighted if >3.27 (upper tertile)); IL-6: Interleukin-6 (highlighted if >52.5 (upper tertile)); As/Mild: asymptomatic or mild.

## Data Availability

The datasets used and/or analyzed during the current study are available from the corresponding authors on reasonable request. No proprietary R codes were used for the purpose of this study and all codes are retrievable online for free.

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
