# Peer review of "The FLARE Score and Circulating Neutrophils in Patients with Cancer and COVID-19 Disease"

_cancers, 2024, doi:10.3390/cancers16172974_

Round 1

Reviewer 1 Report

Comments and Suggestions for Authors

Dear the Editor
Segui E et al reported the usefulness of FLARE score based on the inflammatory conditions induced by either covid19 (I) and tumor(T). These authors reported that the group of I+/T+ led to worst probability of suvival (Fig. 2; Table 1B). An alteration of IL-6 was correlated with clinical stage (baseline/clinical deterioration/recovery) (Fig. 4).

Major concerns:
1) The definition of I+/T+ seemed narrative (LL243-253). The definition of FLARE appeared to be obscure in this manuscript.

Author Response

Thank you very much for taking the time to review this manuscript. We are pleased to submit the revised version of the manuscript for publication. Please find the detailed responses in the attached document and the corresponding revisions/corrections highlighted in the re-submitted manuscript.

Reviewer 2 Report

Comments and Suggestions for Authors

This manuscript titled as “The FLARE score and circulating neutrophils in patients with cancer and Covid-19 disease” attemptes to explore the relationship between cancer-induced inflammation and Covid-19 outcomes through the development of the FLARE score, involving a retrospective cohort of 524 patients and a prospective exploratory cohort. Authors suggested that pre-existing tumor-related inflammation exacerbates Covid-19 severity, and introduced the FLARE score as a predictive tool. However, several critical issues need to be addressed for the manuscript to be suitable for publication. This study proposes an interesting hypothesis and offers preliminary insights, but lacks the depth and breadth required for conclusive evidence. Also, content of this study lacks novelty and clinical significance. Major revisions are recommended.

1.Authors claimed that they identified the impact of cancer-induced inflammation on Covid-19 outcomes through the FLARE score. However, specific signaling pathways or molecular mechanisms were not clearly elucidated. The manuscript should provide detailed descriptions and confirmatory methods for these pathways.

2.The study discusses various cancer subtypes but lacks stratified analysis based on these subtypes. Given the heterogeneous nature of cancer, especially in its inflammatory responses, a stratified analysis by cancer type could provide more precise and clinically relevant insights.

3.The results section needs more detailed descriptions of the findings. For instance, in the sections discussing the development and validation of the FLARE score, the authors should include specific data points, statistical analyses, and outcomes for each subgroup within the score categories.

4.The manuscript does not adequately discuss the practical significance and potential clinical applications of the FLARE score. The sections on "abstract," "discussion," and "conclusions" should be expanded to clearly articulate the novel contributions of this research and its implications for patient care.

5.The authors concluded that a decreased expression of genes related to immune processes is involved in cancer-related inflammation affecting Covid-19 outcomes. However, specific genes and their roles were not clearly identified or confirmed experimentally. These details should be included to support the conclusions drawn.

6.The limitations section briefly mentions the retrospective nature of the study and the small size of the prospective cohort. This section should be expanded to discuss these and other limitations in greater detail. Additionally, the need for future studies to validate and extend these findings should be emphasized, particularly those involving larger and more diverse patient populations.

Author Response

(The authors gave the same response as above.)

Round 2

Reviewer 1 Report

Comments and Suggestions for Authors

Dear the Editor

All raised concerns appeared to be properly cleared.

Reviewer 2 Report

Comments and Suggestions for Authors

This manuscript advice the development of the FLARE score as a predictive tool for assessing the risk of severe Covid-19 complications and mortality in cancer patients is innovative. This score, which incorporates both tumor and infection-induced inflammation markers, offers a new way to stratify patient risk, potentially guiding more effective clinical management. By highlighting the role of inflammatory markers such as the derived neutrophil-to-lymphocyte ratio (dNLR) and IL-6, the article emphasizes the importance of the inflammatory response in cancer and Covid-19. After revision, acceptance can be considered.